# Fabrication and Characterization of Magnetic Cellulose–Chitosan–Alginate Composite Hydrogel Bead Bio-Sorbent

**DOI:** 10.3390/polym15112494

**Published:** 2023-05-29

**Authors:** Aida Syafiqah Abdul Rahman, Ahmad Noor Syimir Fizal, Nor Afifah Khalil, Ahmad Naim Ahmad Yahaya, Md. Sohrab Hossain, Muzafar Zulkifli

**Affiliations:** 1Universiti Kuala Lumpur, Branch Campus Malaysian Institute of Chemical and BioEngineering Technology, 78000 Alor Gajah, Melaka, Malaysia; aida.syafiqah@s.unikl.edu.my (A.S.A.R.); nafifah.khalil@s.unikl.edu.my (N.A.K.); 2Centre for Sustainability of Ecosystem & Earth Resources (Pusat ALAM) Universiti Malaysia Pahang, Lebuh Persiaran Tun Khalil Yaakob, 26300 Gambang, Pahang, Malaysia; syimir@umpholdings.my; 3Green Chemistry and Sustainability Cluster, Universiti Kuala Lumpur, Branch Campus Malaysian Institute of Chemical and BioEngineering Technology, 78000 Alor Gajah, Melaka, Malaysia; ahmadnaim@unikl.edu.my; 4HICoE-Centre for Biofuel and Biochemical Research, Institute of Self-Sustainable Building, Fundamental and Applied Sciences Department, Universiti Teknologi PETRONAS (UTP), 32610 Seri Iskandar, Perak, Malaysia; sohrab.hossain@utp.edu.my

**Keywords:** cellulose, chitosan, alginate, magnetite, bio-sorbent, hydrogel beads, adsorption, morphology

## Abstract

The implementation of inorganic adsorbents for the removal of heavy metals from industrial effluents generates secondary waste. Therefore, scientists and environmentalists are looking for environmentally friendly adsorbents isolated from biobased materials for the efficient removal of heavy metals from industrial effluents. This study aimed to fabricate and characterize an environmentally friendly composite bio-sorbent as an initiative toward greener environmental remediation technology. The properties of cellulose, chitosan, magnetite, and alginate were exploited to fabricate a composite hydrogel bead. The cross linking and encapsulation of cellulose, chitosan, alginate, and magnetite in hydrogel beads were successfully conducted through a facile method without any chemicals used during the synthesis. Energy-dispersive X-ray analysis verified the presence of element signals of N, Ca, and Fe on the surface of the composite bio-sorbents. The appearance and peak’s shifting at 3330–3060 cm^−1^ in the Fourier transform infrared spectroscopy analysis of the composite cellulose–magnetite–alginate, chitosan–magnetite–alginate, and cellulose–chitosan–magnetite–alginate suggested that there are overlaps of O-H and N-H and weak interaction of hydrogen bonding with the Fe_3_O_4_ particles. Material degradation, % mass loss, and thermal stability of the material and synthesized composite hydrogel beads were determined through thermogravimetric analysis. The onset temperature of the composite cellulose–magnetite–alginate, chitosan–magnetite–alginate, and cellulose–chitosan–magnetite–alginate hydrogel beads were observed to be lower compared to raw-material cellulose and chitosan, which could be due to the formation of weak hydrogen bonding resulting from the addition of magnetite Fe_3_O_4_. The higher mass residual of cellulose–magnetite–alginate (33.46%), chitosan–magnetite–alginate (37.09%), and cellulose–chitosan–magnetite–alginate (34.40%) compared to cellulose (10.94%) and chitosan (30.82%) after degradation at a temperature of 700 °C shows that the synthesized composite hydrogel beads possess better thermal stability, owing to the addition of magnetite and the encapsulation in the alginate hydrogel beads.

## 1. Introduction

Heavy metals have been identified as prominent hazardous contaminants in industrial wastewater and water streams due to their non-biodegradable nature, leading to bioaccumulation in biological systems [1]. The presence of heavy metal contamination is a significant concern due to their toxicity and carcinogenic properties, posing multiple health risks to both humans and the environment [2]. The electroplating industry is one of the contributors to the high levels of heavy metal wastewater, where an electro-deposition process is commonly used for coating metals to prevent corrosion [3,4].

The adsorption process is widely employed as a common and effective method for removing heavy metals from wastewater due to its simplicity and efficiency [5]. Adsorption involves reversible attraction between particles, where a solid surface (adsorbent) removes a specific compound (adsorbate) from wastewater. Desorption, on the other hand, releases contaminants from the adsorbent, rendering it reusable for subsequent adsorption cycles [6]. Anionic polyacrylamide polymer (PAAM) is a commonly used synthetic polymer in the coagulation–flocculation treatment process for industrial wastewater [7]. Although the coagulation–flocculation process exhibits excellent treatment efficiency and ease of operation, it introduces a new environmental issue related to the generation and disposal of sludge waste [8,9]. Furthermore, the use of synthetic polymers in conventional industrial wastewater treatment contributes to environmental problems due to the presence of acrylamide, a carcinogenic monomer formed during degradation [10].

To address these challenges, it is crucial to implement alternative environmentally friendly materials, especially for the removal of heavy metals from industrial effluents. Natural polymers such as cellulose, chitosan, and alginate have shown remarkable properties and affinity towards heavy metal ions [11,12]. Several composite adsorbents have been developed for heavy metal removal, including those based on cellulose, chitosan, alginate, activated carbon, and zeolite [13,14]. Previous research has demonstrated the effectiveness of cellulose, chitosan, and magnetite in removing heavy metals, motivating the exploration of these natural and readily available polymers as adsorbents [15,16,17,18,19,20,21]. While chitosan exhibits excellent heavy metal adsorption properties, it has limited mechanical strength, which can be overcome through cross linking with cellulose, leveraging its stiffness to support and reinforce the adsorbent thermally and mechanically [12,15,21,22]. Alginate facilitates the formation of hydrogel beads by cross linking with divalent ions (Ca^2+^), and possesses functional groups that exhibit affinity towards heavy metal cations [12,23]. The combination of the polysaccharide cellulose, chitosan, and alginate as a composite adsorbent addresses their individual stability issues [18]. The addition of magnetite to the composite hydrogel beads enhances the removal efficiency of the adsorbent from the aqueous system by utilizing a magnet, while maintaining the stability of the composite in the hydrogel form [24]. Sodium alginate is employed to encapsulate the composite materials, simplifying the formation of hydrogel beads through the cross-linking process [12,25].

The composite hydrogel bead bio-sorbent developed in this study is a cellulose–chitosan–magnetite–alginate composite hydrogel bead bio-sorbent (CCMA). The focus of this study is to fabricate and characterize this (CCMA) composite hydrogel bead bio-sorbent. This study aimed to address the issue by developing an environmentally friendly bio-sorbent through the encapsulation and cross linking of cellulose, chitosan, and magnetite Fe**_3_**O**_4_** in the hydrogel beads of an alginate polymer matrix. By utilizing cellulose, chitosan, and magnetite encapsulation in alginate hydrogel beads, the current bio-sorbent aims to provide a more convenient and environmentally friendly alternative for heavy metal removal in wastewater.

## 2. Materials and Methods

### 2.1. Materials

Sodium alginate powder (chemically pure) and cellulose powder (chemically pure) were supplied by R&M Chemicals. Chitosan powder (de-acetylation degree of >90% and MW: 10~20 kDa) was supplied by Bio Basic. Magnetite iron (II, III) oxide powder (97% trace metals and MW: 231.53 g/mol) and chloride anhydrous powder (chemically pure and MW: 147.01 g/mol) were supplied by Sigma Aldrich (Saint Louis, MO, USA) and Merck (Rahway, NJ, USA), respectively. All chemicals used were of analytical grade without any modification.

### 2.2. Synthesis of Composite Hydrogel Bead Bio-Sorbents

The composite hydrogel bead bio-sorbents cellulose–magnetite–alginate (CeMA), chitosan–magnetite–alginate (CMA), and cellulose–chitosan–magnetite–alginate (CCMA) in this study were synthesized through a facile cross-linking and encapsulation method. To begin, a sodium alginate solution was prepared by combining 4 g sodium alginate powder with 200 mL ultrapure water under consistent stirring for 1 h at room temperature. Next, cellulose, chitosan, and magnetite powder were, respectively, added into the sodium alginate solution to produce CeMA, CMA, and CCMA bio-sorbent solutions. For CeMA, 2 g of cellulose powder and 0.2 g of magnetite iron (II, III) oxide powder were added into the alginate solution before undergoing stirring for 2 h at 570 rpm and 45 °C. Then, for CMA, 2 g of chitosan powder and 0.2 g of magnetite iron (II, III) oxide powder were added into the alginate solution before undergoing the same stirring condition. Meanwhile, for CCMA, 2 g of cellulose powder and 0.4 g of magnetite iron (II, III) oxide powder solution were added into the alginate solution and it was stirred for 2 h at 570 rpm and 45 °C. Then, 2 g of chitosan powder was added to the solution and it was further stirred for another 2 h under the same stirring condition. Next, for the formation of composite hydrogel beads, CeMA, CMA, and CCMA bio-sorbent solutions were, respectively, dropped into a separate beaker of 100 mL of 0.2 M calcium chloride solution using a burette at a constant speed with approximately 10 cm distance from the solution. The hydrogel beads were formed instantly and were allowed to cure in the calcium chloride solution for 24 h at room temperature. To remove excess calcium chloride, the hydrogel beads collected were rinsed with deionized water multiple times before being stored in the deionized water for further use and analysis.

### 2.3. Characterization Test of the Hydrogel Beads

The morphology and structure of CeMA, CMA, and CCMA composite hydrogel bead samples were identified through scanning electron microscopy (SEM) and energy-dispersive X-ray (EDX). The samples were analyzed through SEM and EDX to identify the changes in the morphology and the elemental analysis of the composite hydrogel beads. All samples were dried out beforehand using a cold vacuum to remove the moisture trapped in the hydrogel beads while retaining their shape. Samples were coated with a thin layer of gold before morphology images were taken using an acceleration voltage of 10 kV. ATR-FTIR (Thermo Scientific, Model: Nicolet iS10, Waltham, MA, USA) was used to determine the functional groups and chemical bonds that exist in CeMA, CMA, CCMA, alginate, cellulose powder, and chitosan powder. A frequency range of spectrophotometer between 4000 and 400 cm^−1^ was applied. The thermal stability of CeMA, CMA, CCMA, alginate, cellulose powder, and chitosan powder was determined using TGA (Mettler Toledo, Zurich, Switzerland). All samples were dried at 50 °C in a drying oven overnight beforehand. Zero-weight calibration was also conducted before analysis. Approximately 10–15 mg of the samples was placed on the pan and heated with nitrogen gas purge from 30 to 700 °C at a heating rate of 10 °C per minute.

## 3. Result and Discussion

### 3.1. Synthesis and Characterization of CeMA, CMA, and CCMA

The synthesis of the composite hydrogel bead bio-sorbents CeMA, CMA, and CCMA were conducted through a facile encapsulation and cross-linking method as described in the Materials and Methods section. Table 1 below shows the materials present in the formulation of CeMA, CMA, and CCMA.

The moisture and mass content of materials present in individual CeMA, CMA, and CCMA composite hydrogel beads are shown in Appendix A. The mass ratio of cellulose/magnetite/alginate in CeMA was 1:0.1:2. For CMA, the mass ratio of chitosan/magnetite/alginate was 1:0.1:2, whereas, for CCMA, the mass ratio of cellulose/chitosan/magnetite/alginate was 1:1:0.1:2. Based on the calculation made (Appendix A), the individual mass content of cellulose, chitosan, magnetite, and alginate in the CeMA and CMA were 0.67 mg, 0.67 mg, 0.067 mg, and 1.33 mg, respectively, while for CCMA, the individual mass content of cellulose, chitosan, magnetite, and alginate was 0.67 mg, 0.67 mg, 0.13 mg, and 1.33 mg, respectively.

On the other hand, the wet weight, dry weight, and moisture content were determined through the moisture analyzer at a drying temperature of 105 °C. It is worth noting that the weight of the dry hydrogel bead sample is also the reflection of the total mass content of the materials in the hydrogel bead sample. All hydrogel beads possess a high moisture content due to the characteristic of hydrogel itself, which is its high-water absorption capacity [25]. Therefore, the high wet weight compared to the dry weight of the hydrogel samples could be contributed by the water content. The weight of CCMA is noticeably higher than that of CeMA and CMA, possibly due to the addition of materials (cellulose and chitosan) in the development of CCMA.

In conclusion, there was only a small number of materials (cellulose, chitosan, magnetite, alginate) utilized for the development of one hydrogel bead with a dry weight of 2.80 mg (CeMA and CMA) and 4.00 mg (CCMA). Moreover, the difference obtained in the total mass via the calculation method and the weight of the dry hydrogel bead could be due to the other material (Ca) that is present in the hydrogel bead during the formulation.

### 3.2. SEM Analysis

Figure 1 shows that the hydrogel beads were spherical with a rough surface. The synthesized CeMA, CMA, and CCMA were spherical shapes with average diameters of 3.51–3.95 mm. However, Zhu et al. [26] reported that the diameter of synthesized magnetic alginate hydrogel beads in a wet state was around 3.41 mm. This demonstrates that the method chosen for the synthesis of the composite hydrogel bead adsorbent can produce hydrogel beads with a consistent diameter and shape. Aside from that, a SEM analysis of cellulose, chitosan, magnetite, and alginate hydrogel beads was performed and is depicted in Figure 2. Furthermore, as shown in Figure 3, the surface of CeMA, CMA, and CCMA appears rough, with many solid particles evenly distributed on it. The encapsulated cellulose, chitosan, and magnetite in the alginate polymer matrix could contribute to these solid particles.

The straight crystalline with a rod-like shape of cellulose in Figure 2a can be seen on CeMA and CCMA’s surface in Figure 3a,e [27]. Additionally, based on Figure 2b, chitosan with a flaky appearance was also observed on the surface of CMA and CCMA in Figure 3c,e [28]. On the other hand, the SEM image of the magnetite particle in Figure 2c shows the agglomeration of the magnetite Fe_3_O_4_. The particle agglomeration is due to the strong Van der Waals forces between the particles [11]. The observations on the magnetite dispersion on the surface of CeMA, CMA, and CCMA were further shown by SEM in back-scattered electron (BSE) mode in Figure 3b,d,f. Furthermore, Figure 2d shows a smooth surface of the alginate hydrogel bead with no apparent solid particles on its surface. By comparing it with Figure 2d, the rougher surface seen in CeMA, CMA, and CCMA is contributed by the addition of cellulose, chitosan, and magnetite in the alginate polymer matrix. Similarly, Germanos et al. (2020) also reported the same finding on the changes in the surface of an alginate bead upon the addition of magnetite, where a rough and irregular alginate bead surface can be seen through SEM image after the addition of magnetite in the alginate polymer matrix [29].

In conclusion, cellulose, chitosan, and magnetite were successfully encapsulated in the alginate polymer matrix. Moreover, the magnetite particles were evenly distributed across the CeMA, CMA, and CCMA surfaces, as shown in the SEM in BSE mode. This proves that the synthesized method used for the development of composite CeMA, CMA, and CCMA successfully produced a well-distributed material on the surface of the hydrogel bead. Moreover, the absence of a porous structure and openings in CeMA, CMA, and CCMA also suggested that the adsorption mechanism of the composite hydrogel bead bio-sorbent does not depend on its physical structure [30].

Figure 4 presents the schematic diagram of the CCMA structure to show the cross linking and encapsulation of the polymers in the composite hydrogel beads. The formation of the 3D-shaped hydrogel beads was contributed by the cross linking of alginate and Ca^2+^ [14]. By the encapsulation of cellulose, chitosan, and magnetite in the alginate polymer matrix, the individual properties of each material can be exploited in a much easier and environmentally friendly way.

### 3.3. EDX Analysis

The existence of cellulose, chitosan, magnetite, and alginate in CeMA, CMA, and CCMA were further proven by the energy-dispersive X-ray analysis (EDX), where the element compositions in the CeMA, CMA, and CCMA were analyzed and discussed.

The elements analyses on the surface of the CeMA, CMA, and CCMA were further studied through the EDX analysis, as shown in Figure 5. Based on Figure 5a, the EDX analysis shows the elements present in CeMA. The signal for Ca is contributed by the cross linking of alginate and Ca for the formation of the hydrogel bead, whereas the Fe signal shows a successful encapsulation of magnetite Fe_3_O_4_ in the alginate polymer matrix. On the other hand, in Figure 5b, the emergence of nitrogen (N) shows the successful combination of chitosan in CMA. Figure 5c shows the elements’ signals on the surface of CCMA. The N, Fe, and Ca signals show the successful encapsulation of chitosan and magnetite in the alginate polymer matrix [12,26]. In addition, the element Ca in CeMA, CMA, and CCMA also proves that an ion exchange mechanism can occur between Ca and heavy metal ions which would facilitate the removal of heavy metals [17].

### 3.4. FTIR Analysis

The confirmation of the elements is further supported by the FTIR analysis in Figure 6 and Figure 7, where the presence of the chemical bonds in CeMA, CMA, and CCMA are observed and discussed.

Figure 6 and Figure 7 show the FTIR analysis that was conducted to identify the functional groups and changes in chemical bonds present in the cellulose, chitosan, magnetite, alginate hydrogel bead, CeMA, CMA, and CCMA in the wavelengths of 400–4000 cm^−1^. Based on previous studies, the FTIR analysis revealed that the functional groups present in the composite consisting of cellulose, chitosan, alginate, and magnetite are hydroxyls, amines, carboxyls, carbonyls, amides, and ethers (Appendix A).

Based on the FTIR spectra shown in Figure 6, the magnetite shows a strong characteristic peak of the Fe-O bond at 531.04 cm^−1^. Moreover, for the cellulose spectrum, the appearance of a broad and strong peak at 3333.09 cm^−1^ corresponds to the O-H stretching [11]. Meanwhile, the weak peaks at 2904.27 cm^−1^ and 1643.48 cm^−1^ can be assigned to the stretching of C-H and C=O (an asymmetrical stretch of carbonyl groups) [31]. The peak at 1427.27 cm^−1^ is assigned to the bending vibrations of CH_2_ at C_6_, whereas the peak at 1027.82 cm^−1^ is the unique characteristic of C-O-C stretching.

In the chitosan spectrum, the appearance of a broad and strong peak at 3355.96 cm^−1^ indicates the overlapping of the O-H and N-H stretching [5], [32]. On the other hand, the weak stretching of C-H appeared at 2871.55 cm^−1^. The medium peak at 1655.17 cm^−1^ is assigned to the carbonyl compounds from the amide [31,33]. Meanwhile, the peaks at 1590.99 cm^−1^ and 1024.99 cm^−1^ are assigned to the N-H bending vibration from the secondary amide and the C-O-C stretching, respectively [31].

Based on Figure 7, the alginate bead spectrum shows a broad strong peak of O-H stretching at 3291.10 cm^−1^. The peaks at 1593.13 cm^−1^ and 1416.22 cm^−1^ are assigned to the carbonyl compounds from the carboxylate anions in alginate (strong asymmetrical C=O stretching and weak symmetrical C-O stretching) [31]. Additionally, the C-O-C stretch in the alginate beads appeared at the peak of 1025.40 cm^−1^.

For the CeMA spectrum, the peak at 3288.15 cm^−1^ is assigned to the O-H stretching. However, the slight shift in the peaks from 3291.10 cm^−1^ (alginate beads) to 3288.15 cm^−1^ is probably due to the formation of the weak bond between O-H and Fe_3_O_4_ [34]. Moreover, the strong peaks at 1598.34 cm^−1^ and 1417.97 cm^−1^ are assigned to the carbonyl compounds from the carboxylate anions (-COO-) from alginate. However, the slight shift in the peak seen in the CeMA could be due to the increase in C-H groups from the polymer backbone of cellulose as a result of the cellulose addition into the alginate [35]. Meanwhile, the C-O-C stretch in CeMA was observed at 1026.10 cm^−1^.

In the CMA spectrum, the shift observed from 3291.10 cm^−1^ (alginate beads) to 3280.61 cm^−1^ could be speculated to be (i) the overlapping of O-H from alginate and N-H from chitosan [31] or (ii) a weak bond formation of O-H and N-H with the Fe_3_O_4_ [34]. Moreover, a slight shift observed at 1596.21 cm^−1^ and 1416.41 cm^−1^ for the carbonyl compounds from the carboxylate anion groups could be due to the NH^3+^ groups from chitosan interacting with the -COO- (carboxylate anion) groups from alginate to form C=N (imine compound) [5,33]. A unique peak of the C-O-C stretch is observed at 1025.57 cm^−1^, which is almost identical to the C-O-C stretch in the alginate bead.

For the CCMA spectrum, the slight shift observed at 3294.06 cm^−1^ could be speculated to be (i) the overlapping of the O-H stretch and N-H stretch from chitosan [32,36] or (ii) the formation of a weak bond between O-H and N-H with the Fe_3_O_4_ particles [34]. In addition, the shift observed at 1595.52 cm^−1^ and 1417.02 cm^−1^ assigned to the carbonyl compounds could be: (i) the interaction of NH^3+^ groups from chitosan and the -COO- (carboxylate anion) groups from alginate to form C=N (imine compound) [5,33] or (ii) an increase in C-H groups from the cellulose polymer backbone [35]. Additionally, a unique peak of the C-O-C stretch was observed at 1025.97 cm^−1^, which is almost identical to the C-O-C stretch seen in alginate beads.

Interestingly, by referring to Figure 7, all peaks in the synthesized hydrogel beads resembled one another, with multiple overlapping peaks. However, the slight shift in the peaks and intensity indicates that there are multiple interactions (electrostatic interaction and hydrogen bonding) between the polymers [31]. Moreover, the absence of the Fe-O bond in CeMA, CMA, and CCMA indicates that the magnetite Fe_3_O_4_ only forms a weak bond on the synthesized composite hydrogel beads [34]. Despite that, the presence of magnetite on the surface of the composite hydrogel beads CeMA, CMA, and CCMA was proven through the SEM-EDX analysis conducted in the study.

### 3.5. TGA

Figure 8 and Figure 9 show the thermal degradation of the raw materials cellulose, chitosan, alginate beads, and magnetite. The major weight loss for cellulose was observed at 80.55% in the second degradation step at a temperature of 300–450 °C, whereas a major weight loss of 51.23% was observed in chitosan at temperatures 230–450 °C in the second degradation step. Additionally, this is on par with the study by Khalid et al. (2021), where the onset degradation temperature of cellulose was observed at a temperature of 316.70 °C [11]. Moreover, Karzar Jeddi and Mahkam (2019) also reported that the weight loss at a temperature range of 270–450 °C was contributed by the degradation of chitosan [31]. Moreover, a major weight loss of 41.42% for alginate beads was observed at the second degradation step in the temperature range of 200–300 °C. In addition, the degradation of magnetite, as seen in Figure 9, also occurred in the second degradation step, where a major weight loss of 3.34% was observed in magnetite in the temperature range of 200–400 °C.

Figure 8 further depicted the TGA curve of CeMA, CMA, and CCMA. From the TGA curve, a slight initial weight loss occurred in all samples at the first degradation step at 30–200 °C due to moisture loss from the residual water in the samples [18,27]. A major weight loss (inflection point) was observed in CeMA, CMA, and CCMA at the second degradation step at 200–400 °C, of 48.45%, 36.68%, and 51.01% respectively. This indicates that the alginate, magnetite, cellulose, and chitosan in the composite hydrogel beads undergo thermal degradation. As claimed by Peng et al. (2017), the degradation of organic compounds and the breakage of functional groups occurred at temperatures above 200 °C [15]. It was observed that the weight loss in CCMA was slightly higher, which could be due to the presence of both cellulose and chitosan in the composite hydrogel bead.

Moreover, breaking the O-H groups in the alginate structure of CeMA, CMA, and CCMA in the temperature range of 200–300 °C also contributed to the second degradation step, as reported by Karzar Jeddi and Mahkam (2019) [31]. Interestingly, the TGA curve showed that all synthesized composite hydrogel bead CeMA, CMA, and CCMA possess better thermal properties than cellulose and chitosan powder due to the addition of magnetite. However, the onset degradation temperature of CeMA, CMA, and CCMA was slightly lower than that of the cellulose and chitosan due to the weaker hydrogen bonding resulting from the interaction with magnetite in CeMA, CMA, and CCMA [11]. Nevertheless, the residual mass of CeMA, CMA, and CCMA after degradation was higher. The final residual mass of CeMA, CMA, and CCMA was observed to be 33.46%, 37.09%, and 34.40%, respectively, as shown in Table 2. Overall, the cross linking of alginate and the addition of magnetite in the development of the CeMA, CMA, and CCMA was proven to improve their thermal properties, compared to the pure cellulose and chitosan.

## 4. Conclusions

The cross linking and encapsulation of cellulose, chitosan, alginate, and magnetite in hydrogel beads were successfully conducted in this study through a facile method without using any chemicals during the synthesis, which facilitates an environmentally friendly material to be used as a bio-sorbent. The synthesized composite bio-sorbents (CeMA, CMA, and CCMA) were proven to have a surface morphology with evenly distributed materials (cellulose, chitosan, alginate, and magnetite). Additionally, the N, Ca, and Fe element signals from the EDX results supported the presence of the materials on the surface of the composite bio-sorbents. The analysis results obtained from the FTIR spectra also supported the presence of cellulose, chitosan, alginate, and magnetite through the existence of functional groups and chemical bonds. Furthermore, the FTIR spectra of the synthesized composite hydrogel beads resembled one another, where the changes and the shift in the intensity and peaks of the spectra showed that there is the possibility of functional groups overlapping and bond formations. The existence and the slight shift in the carbonyl compounds from the carboxylate anions were contributed by alginate. A TGA analysis showed that the biosorbent % mass loss observed at the degradation temperature range of 30–200 °C contributed to the moisture loss in the samples. The onset temperature of CeMA, CMA, and CCMA is lower than that of cellulose and chitosan due to the formation of weak hydrogen bonds resulting from the addition of magnetite Fe_3_O_4_. A higher residual mass of CeMA (33.46%), CMA (37.09%), and CCMA (34.40%) was observed in comparison to cellulose (10.94%) and chitosan (30.82%) after degradation at a temperature of 700 °C. This indicates that the synthesized composite hydrogel beads have better thermal stability due to the addition of magnetite and the encapsulation in the alginate hydrogel beads. The synthesized environmentally friendly composite bio-sorbent characteristics were shown to be desirable for heavy metal wastewater treatment by having functional groups with a good affinity towards heavy metals and improved thermal stability compared to the raw cellulose and chitosan.

## Figures and Tables

**Figure 1 polymers-15-02494-f001:**
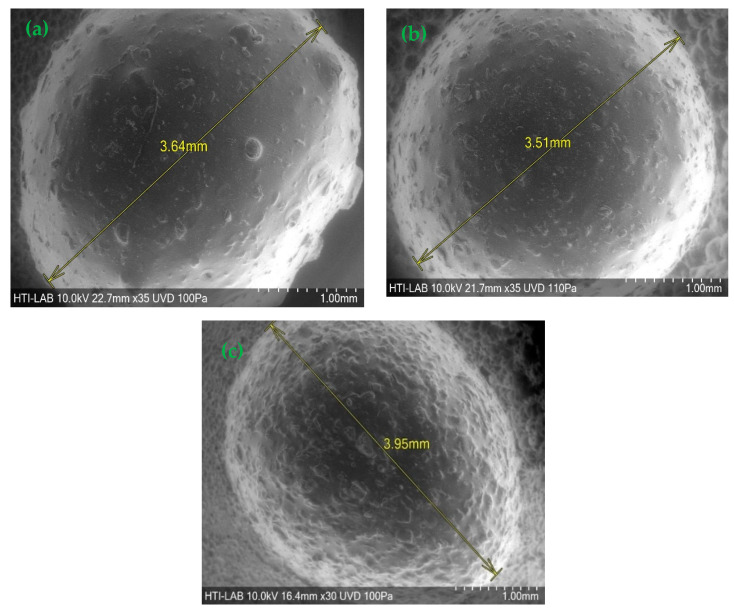
SEM image of (**a**) CeMA, (**b**) CMA, and (**c**) CCMA.

**Figure 2 polymers-15-02494-f002:**
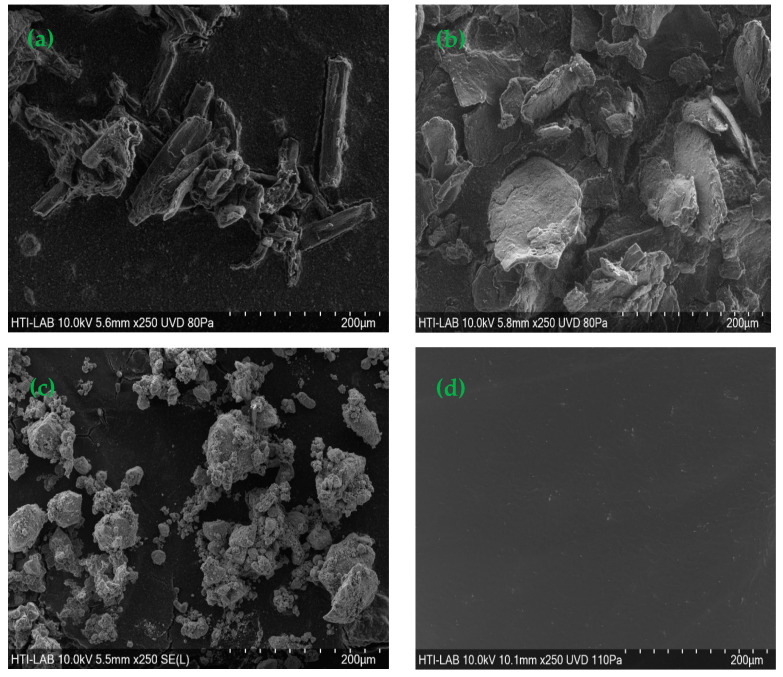
SEM image of (**a**) cellulose, (**b**) chitosan, (**c**) magnetite Fe_3_O_4_, and (**d**) alginate hydrogel bead.

**Figure 3 polymers-15-02494-f003:**
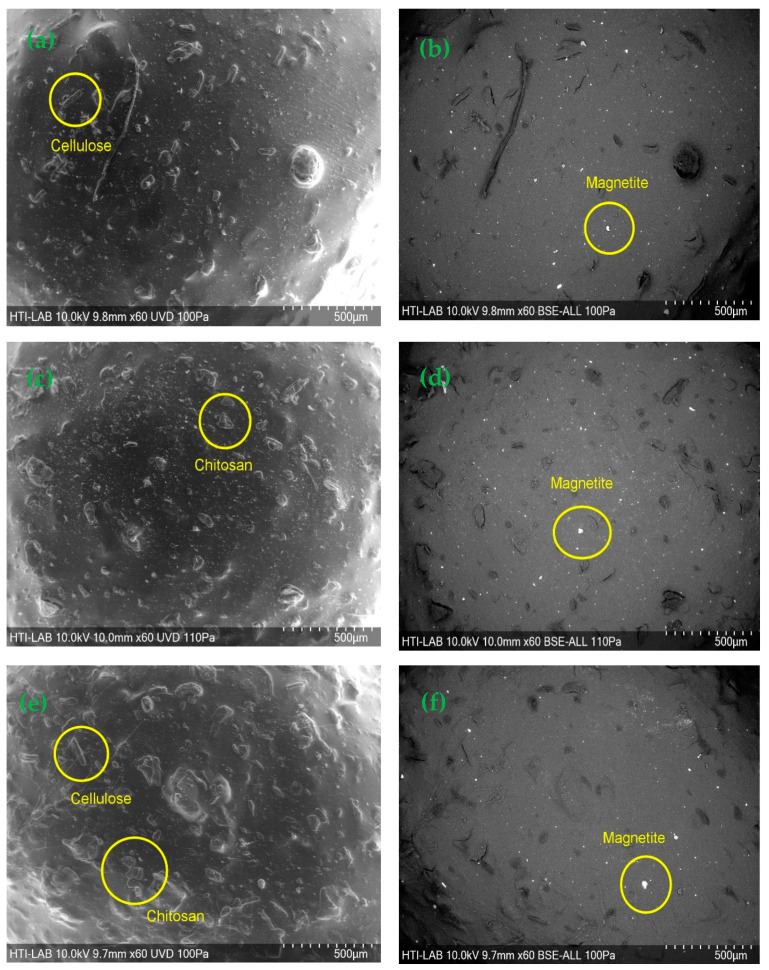
SEM image of (**a**) CeMA, (**b**) CeMA in BSE mode, (**c**) CMA, (**d**) CMA in BSE mode, (**e**) CCMA, and (**f**) CCMA in BSE mode.

**Figure 4 polymers-15-02494-f004:**
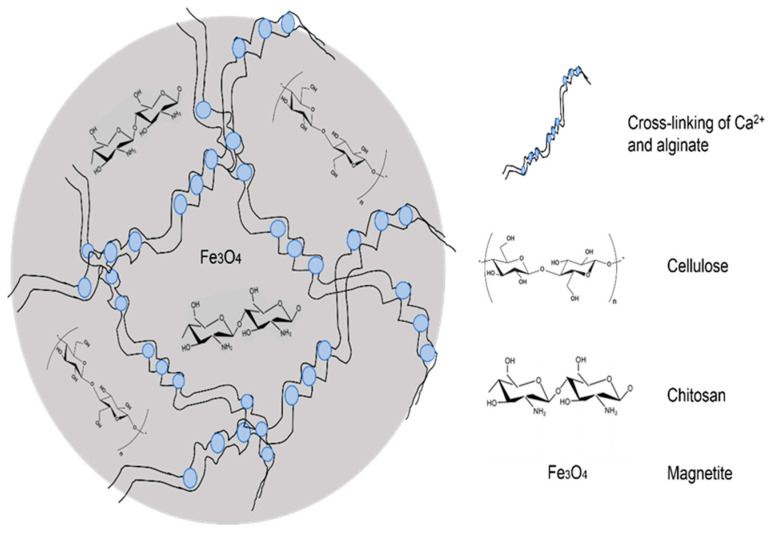
Schematic diagram of CCMA structure.

**Figure 5 polymers-15-02494-f005:**
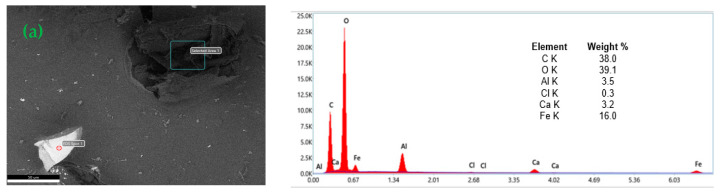
Energy-dispersive X-ray (EDX) analysis of CeMA (**a**), CMA (**b**), and CCMA (**c**).

**Figure 6 polymers-15-02494-f006:**
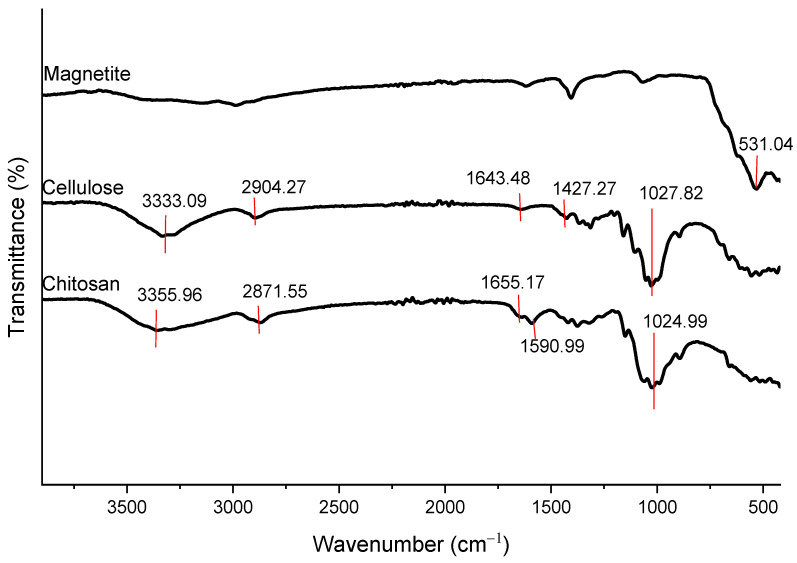
FTIR of magnetite, cellulose, and chitosan.

**Figure 7 polymers-15-02494-f007:**
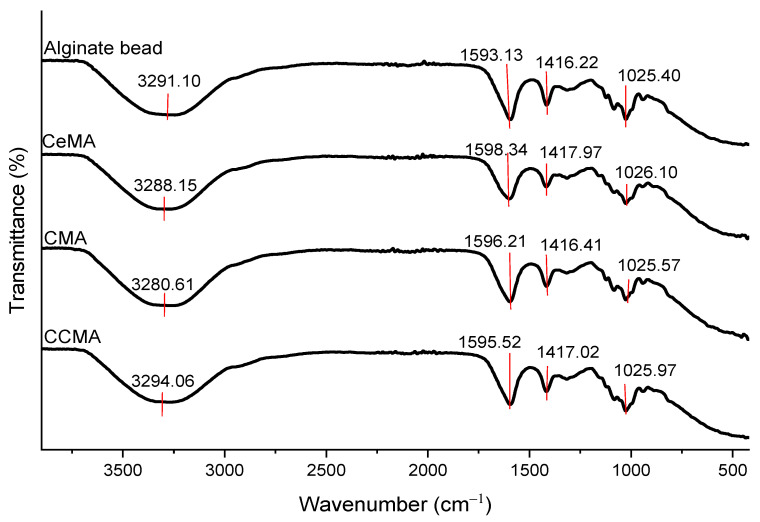
FTIR of alginate beads, CeMA, CMA, and CCMA.

**Figure 8 polymers-15-02494-f008:**
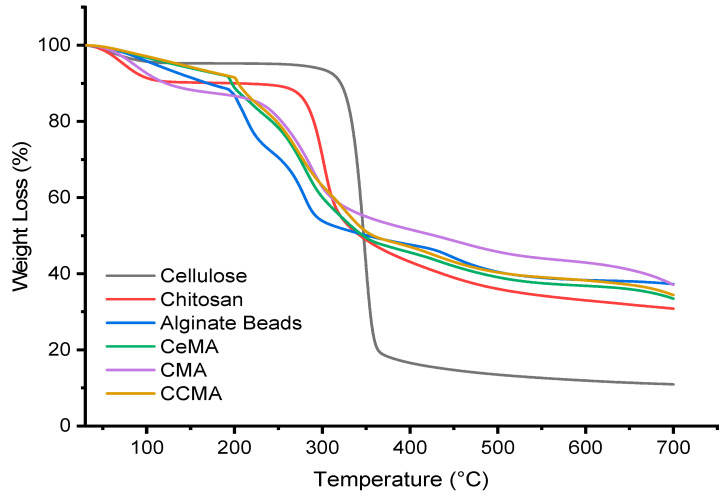
TGA curve of cellulose, chitosan, alginate, CeMA, CMA, and CCMA.

**Figure 9 polymers-15-02494-f009:**
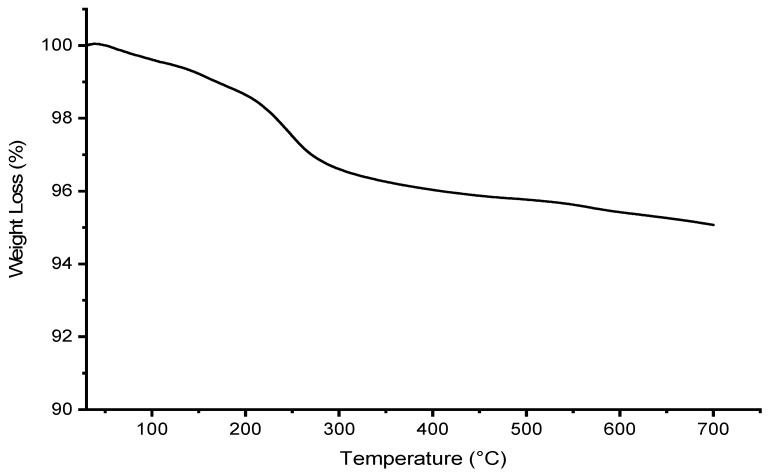
TGA curve of magnetite.

**Table 1 polymers-15-02494-t001:** Materials present in CeMA, CMA, and CCMA.

Hydrogel Bead	Cellulose	Chitosan	Alginate	Magnetite	Mass Ratio of Cellulose/Chitosan/Magnetite/Alginate
CeMA	√	X	√	√	1:0:0.1:2
CMA	X	√	√	√	0:1:0.1:2
CCMA	√	√	√	√	1:1:0.1:2

**Table 2 polymers-15-02494-t002:** Thermogravimetric analysis of cellulose, chitosan, magnetite, alginate hydrogel beads, CeMA, CMA, and CCMA.

Sample	Mass Ratio of Cellulose/Chitosan/Magnetite/Alginate	Inflection Point °C	Weight Loss at Inflection Point (%)	Mass Residual (%) at 700 °C
Cellulose	NA	347.74	80.55	10.94
Chitosan	NA	304.03	51.23	30.82
Magnetite Fe_3_O_4_	NA	250.00	3.34	95.07
Alginate hydrogel bead	NA	248.35	41.42	37.27
CeMA	1:0:0.1:2	274.51	48.45	33.46
CMA	0:1:0.1:2	282.30	36.68	37.09
CCMA	1:1:0.1:2	284.16	51.01	34.40

## Data Availability

The data presented in this study are available on request from the corresponding author.

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
