# Peer review of "Fabrication and Characterization of Magnetic Cellulose–Chitosan–Alginate Composite Hydrogel Bead Bio-Sorbent"

_polymers, 2023, doi:10.3390/polym15112494_

Round 1

Reviewer 1 Report

Aida Syafiqah Abdul Rahman et al. reported “Fabrication and Characterization of Magnetic Cellulose–Chitosan–Alginate Composite Hydrogel Beads Bio-Sorbent”. The present work is not publishable in the present form and consider after addressing the following major issues.

1.       Remove the useless keywords like FTIR, TGA. Have you ever seen such keywords in the literature?

2.       Replace the abbreviations with full form in the abstract and abstract need careful revision.

3.       The introduction part need to be re-write with expert. The present introduction is looking very rough.

4.       XRD and TEM analysis should be included in the revised manuscript for the confirmation iron nanoparticles.

5.       In Table 1 the exact amount or ratio of the used Materials should be included.

6.       FTIR need to be re-draw in origin software again because the present one is looking images.

7.       Application should be included in the revised manuscript. The present study is very simple without application. Before acceptance application should be included.

8.       The possible interaction mechanism of the used material should be included in the form of schematic representation that what kind of interaction occurs during this process.

9.       Extensive English revision is needed and paper need to be revised by expert with additional experiment and characterization.

Improve the English through expert.

Author Response

Dear reviewer,

Thank you for the comments and suggestions for the improvement of this manuscript. Please see attached the revised manuscript and the response to the reviewer. 

Reviewer 2 Report

Dear Respectful Editor,

Thank you for giving me the opportunity to review this review article titled "Fabrication and Characterization of Magnetic Cellulose–Chitosan–Alginate Composite Hydrogel Beads Bio-Sorbent" and would like to provide my feedback.

The study aimed to develop an environmentally friendly composite bio-sorbent for the removal of heavy metals from industrial effluent. The authors successfully cross-linked and encapsulated Cellulose, chitosan, alginate, and magnetite in hydrogel beads using a facile method without any chemical used during the synthesis. The synthesized composite hydrogel beads were characterized using various techniques such as EDX, FTIR, and TGA. The methodology and results presented in the study appear to be sound, and the study has the potential to make a valuable contribution to the field of hydrogel synthesis.

However, there are some issues that need to be addressed in order to improve the clarity and quality of the article. Below are some specific comments and suggestions for improvement:

·      I would like to suggest some small changes to the the article to improve its clarity and accuracy.

For example: The properties of Cellulose, chitosan, magnetite, and alginate were exploited as a composite hydrogel bead (CCMA). 

The properties of cellulose, chitosan, magnetite, and alginate were exploited to fabricate a composite hydrogel bead (CCMA).

For example: A few common composite adsorbents developed for heavy metal removal are cellulose-based adsorbent, chitosan-based adsorbent, alginate-based adsorbent, activated carbon-based adsorbent, and zeolite-based adsorbent [13-14]. 

Several composite adsorbents have been developed for heavy metal removal, including those based on cellulose, chitosan, alginate, activated carbon, and zeolite [13-14].

For example: This study also aimed to solve the problem by developing an environmentally friendly bio-sorbent through encapsulation and cross-linking of cellulose, chitosan, and magnetite Fe3O4 in the alginate polymer matrix hydrogel beads bio-sorbent. 

This study aimed to address the issue by developing an environmentally friendly bio-sorbent through the encapsulation and cross-linking of cellulose, chitosan, and magnetite Fe3O4 in the hydrogel beads of an alginate polymer matrix.

·      Please provide information on the purity and molecular weight of the chemicals.

In the synthesis part, there are some sentences that are very similar. To make the summary more understandable, we can rephrase it by combining similar sentences and removing any unnecessary repetition.

·      The correct format for the unit cm-1 is with a superscript "−1", as in "cm^−1".

·      Please provide a clearer version of Figure 2d for better visibility and understanding.

·      Please provide a separate TGA graph for magnetite.

Although the use of hydrogels as adsorbents was suggested in the text, no experimental data on the adsorption process were presented. Therefore, in order to fully evaluate the potential of the synthesized composite hydrogel beads as bio-sorbents, swelling, kinetic, isotherm, and thermodynamic calculations should be carried out and presented. Without these data, the study is limited to synthesis and characterization work only.

Best regards,

Dear Respectful Editor,

Thank you for giving me the opportunity to review this review article titled "Fabrication and Characterization of Magnetic Cellulose–Chitosan–Alginate Composite Hydrogel Beads Bio-Sorbent" and would like to provide my feedback.

The study aimed to develop an environmentally friendly composite bio-sorbent for the removal of heavy metals from industrial effluent. The authors successfully cross-linked and encapsulated Cellulose, chitosan, alginate, and magnetite in hydrogel beads using a facile method without any chemical used during the synthesis. The synthesized composite hydrogel beads were characterized using various techniques such as EDX, FTIR, and TGA. The methodology and results presented in the study appear to be sound, and the study has the potential to make a valuable contribution to the field of hydrogel synthesis.

However, there are some issues that need to be addressed in order to improve the clarity and quality of the article. Below are some specific comments and suggestions for improvement:

·      I would like to suggest some small changes to the the article to improve its clarity and accuracy.

For example: The properties of Cellulose, chitosan, magnetite, and alginate were exploited as a composite hydrogel bead (CCMA). 

The properties of cellulose, chitosan, magnetite, and alginate were exploited to fabricate a composite hydrogel bead (CCMA).

For example: A few common composite adsorbents developed for heavy metal removal are cellulose-based adsorbent, chitosan-based adsorbent, alginate-based adsorbent, activated carbon-based adsorbent, and zeolite-based adsorbent [13-14]. 

Several composite adsorbents have been developed for heavy metal removal, including those based on cellulose, chitosan, alginate, activated carbon, and zeolite [13-14].

For example: This study also aimed to solve the problem by developing an environmentally friendly bio-sorbent through encapsulation and cross-linking of cellulose, chitosan, and magnetite Fe3O4 in the alginate polymer matrix hydrogel beads bio-sorbent. 

This study aimed to address the issue by developing an environmentally friendly bio-sorbent through the encapsulation and cross-linking of cellulose, chitosan, and magnetite Fe3O4 in the hydrogel beads of an alginate polymer matrix.

·      Please provide information on the purity and molecular weight of the chemicals.

In the synthesis part, there are some sentences that are very similar. To make the summary more understandable, we can rephrase it by combining similar sentences and removing any unnecessary repetition.

·      The correct format for the unit cm-1 is with a superscript "−1", as in "cm^−1".

·      Please provide a clearer version of Figure 2d for better visibility and understanding.

·      Please provide a separate TGA graph for magnetite.

Although the use of hydrogels as adsorbents was suggested in the text, no experimental data on the adsorption process were presented. Therefore, in order to fully evaluate the potential of the synthesized composite hydrogel beads as bio-sorbents, swelling, kinetic, isotherm, and thermodynamic calculations should be carried out and presented. Without these data, the study is limited to synthesis and characterization work only.

Best regards,

Author Response

(The authors gave the same response as above.)

Round 2

Reviewer 1 Report

The author addressed all the issues in the revised manuscript and I am recommend the manuscript for publication in the present form.

Its OK now

Reviewer 2 Report

Dear Editor,

I am writing to provide my decision as a reviewer for the manuscript titled "Fabrication and Characterization of Magnetic Cellulose–Chitosan–Alginate Composite Hydrogel Beads Bio-Sorbent," submitted to Polymers for consideration for publication. Having carefully reviewed the manuscript, I am pleased to recommend its acceptance for publication in Polymers. The authors have conducted a comprehensive study on the fabrication and characterization of magnetic cellulose–chitosan–alginate composite hydrogel beads as a bio-sorbent material. The manuscript presents valuable findings and contributes significantly to the field of hydrogel-based bio-sorbents. The manuscript is well-written, and the methodology is clearly explained. The authors have presented their results and discussion in a structured and organized manner. The figures and tables are also clear and informative. Overall, I find the manuscript to be scientifically sound and well-presented. Therefore, I highly recommend its acceptance for publication in your esteemed journal.

Thank you for considering my recommendation.

Best regards.